# Metabolic Remodeling in Skeletal Muscle Atrophy as a Therapeutic Target

**DOI:** 10.3390/metabo11080517

**Published:** 2021-08-05

**Authors:** Alessandra Renzini, Carles Sánchez Riera, Isidora Minic, Chiara D’Ercole, Biliana Lozanoska-Ochser, Alessia Cedola, Giuseppe Gigli, Viviana Moresi, Luca Madaro

**Affiliations:** 1Unit of Histology and Medical Embryology, Department of Anatomy, Histology, Forensic Medicine and Orthopaedics, Sapienza University of Rome, 00185 Rome, Italy; alessandra.renzini@uniroma1.it (A.R.); carles.sanchezr@gmail.com (C.S.R.); minic.1916003@studenti.uniroma1.it (I.M.); c.dercole@uniroma1.it (C.D.); biliana.lozanoska-ochser@uniroma1.it (B.L.-O.); luca.madaro@uniroma1.it (L.M.); 2Institute of Nanotechnology, c/o Dipartimento di Fisica, National Research Council (CNR-NANOTEC), Sapienza University of Rome, 00185 Rome, Italy; alessia.cedola@cnr.it; 3Institute of Nanotechnology, c/o Campus Ecotekne, National Research Council (CNR-NANOTEC), Monteroni, 73100 Lecce, Italy; giuseppe.gigli@cnr.it

**Keywords:** skeletal muscle metabolism, muscle wasting, physical exercise, diet, epigenetics

## Abstract

Skeletal muscle is a highly responsive tissue, able to remodel its size and metabolism in response to external demand. Muscle fibers can vary from fast glycolytic to slow oxidative, and their frequency in a specific muscle is tightly regulated by fiber maturation, innervation, or external causes. Atrophic conditions, including aging, amyotrophic lateral sclerosis, and cancer-induced cachexia, differ in the causative factors and molecular signaling leading to muscle wasting; nevertheless, all of these conditions are characterized by metabolic remodeling, which contributes to the pathological progression of muscle atrophy. Here, we discuss how changes in muscle metabolism can be used as a therapeutic target and review the evidence in support of nutritional interventions and/or physical exercise as tools for counteracting muscle wasting in atrophic conditions.

## 1. Introduction

Skeletal muscle is composed of heterogenous fibers with different metabolic profiles and functional properties. Muscle fibers are broadly categorized into two groups: the slow-twitch or type 1 fibers and the fast-twitch or type 2 fibers. Because of the differential expression of myosin heavy chains, fast-twitch myofibers are further subdivided into 2A, 2X, and 2B types, and into 1/2A, 2A/2X, and 2X/2B types based on hybrid myosin heavy chain expression [1]. This classification also reflects differences in mitochondrial content and ATP consumption [1,2]. Indeed, type 1 and 2A fibers are primarily oxidative, while type 2X and 2B fibers mainly rely on glycolytic metabolism [1,2]. In addition to metabolism and myosin heavy chain expression, numerous other factors contribute to fiber-type classification, including the expression of specific components of the sarcomere contractile machinery [1,2], the use of alternative splicing for some structural proteins [3,4], or the expression of fiber-specific micro-RNAs [5]. Ultimately, the orchestrated regulation of fiber-type-specific biochemical and physiological features results in unique and specific functional properties for each fiber type. In mammals, a single muscle group is generally formed by multiple fiber types, and different proportions of fiber types lead to diverse muscle groups [1,2]. For instance, a soleus muscle is mainly composed by type 1 slow fibers while a triceps one is predominantly type 2 [6]. Nevertheless, muscle fibers can remodel their phenotypes in response to external demand. Depending on the stimuli, muscle atrophy may affect specific fiber types, involving predominantly slow type I or fast type II muscle fibers [7]. Indeed, muscle fiber subtypes respond differently to distinct signaling pathways. In general, fast-twitch glycolytic fibers are more prone to be affected by atrophic conditions than slow-twitch oxidative fibers [8].

Muscle atrophy is often associated with metabolic reprogramming, which contributes to phenotypical changes and protein degradation. Conditions known to induce changes in muscle metabolism and initiate muscle wasting include aging, denervation, or neurodegenerative diseases, such as amyotrophic lateral sclerosis (ALS), and syndromes, such as cancer cachexia. Compromised muscle homeostasis, due to a decrease in protein synthesis, and perturbed proteolytic pathways are features shared by all of the above conditions; however, differences in muscle metabolic reprogramming have been reported, dependent on different stimuli [7]. For instance, decreased usage of skeletal muscle causes an adaptive metabolic remodeling characterized by lower capacity for fat oxidation and switch to glycolysis as the main fuel supply, associated with a transition from slow to fast myosin fiber types [9]. Instead, increased fatty acid oxidation and glucose intolerance, associated with a shift in fast-to-slow muscle fiber type composition [10], have been reported in a mouse model of ALS [11].

As an active part of the muscle wasting process, metabolic remodeling represents an attractive therapeutic target for the re-establishment of muscle homeostasis. Thus, the use of thyroid hormone prevents fasting-induced skeletal muscle atrophy by inducing metabolic adaptation, without affecting muscle synthesis or degradation [12]. Moreover, physical exercise or specific nutrient intake can counteract skeletal muscle atrophy in different pathophysiological conditions [13,14,15]. In this review, we will give an overview of the involvement of metabolic remodeling in muscle pathophysiology upon different stimuli, as well as the signaling pathways underlying compromised muscle homeostasis. We will discuss how physical exercise or nutritional support can counteract muscle wasting, proposing them as nonpharmacological approaches to restore muscle homeostasis, even in disease contexts.

## 2. Skeletal Muscle Metabolic Reprogramming during Aging

Muscle is a highly diverse and adaptive tissue [16]. However, during aging, changes in metabolism and accumulation of reactive oxygen species (ROS) are associated with loss of muscle mass and decline in muscle function rather than being part of an adaptive response [17] (Figure 1). Loss of muscle mass is proportional to the passage of time, increasing from 1–2% around 50 years of age, to 40% in people older than 80 years who are considered sarcopenic [18,19]. Related to muscle fiber loss, there is a decline in muscle functions, such as temperature regulation [20]. Moreover, there is an imbalance between anabolic and catabolic processes in skeletal muscle, with a decrease in anabolic agents such as insulin-like growth factor (IGF-1), testosterone, and growth hormone (GH), and an increase in catabolic signals such as interleukin 6 (IL-6) and tumor necrosis factor-alpha (TNF-α) [21]. Indeed, sarcopenia is increasingly recognized as an inflammatory state driven by cytokines and oxidative stress [22] that negatively impacts on muscle protein turnover, by decreasing protein synthesis [23].

While in the early stages of development, cell energy (ATP) and oxidative phosphorylation are generated through the oxidation of carbon sources such as lactate, pyruvate, amino acids, and fatty acids [24,25,26]; during aging, muscles are subjected to metabolic changes. For instance, mitochondrial activity impairment and changes in energy metabolism are critical to the development of sarcopenia [27]. Mitochondria in skeletal muscles of the elderly display impaired respiration, a decreased activity of mitochondrial enzymes, and reduced mitochondrial biogenesis and content [28,29]. Thus, oxidative muscles, which are endowed with greater mitochondrial content, are able to preserve their oxidative capacity during aging-associated changes, unlike glycolytic muscles, which are more prone to becoming impaired [30].

Muscle loss in sarcopenia results from a fiber type-specific loss of muscle mass (muscle atrophy) and reduced number of specific fiber types (hypoplasia). Numerous studies have shown that during aging, type II fibers are more susceptible to atrophy compared to type I fibers [31,32]. Moreover, type IIB fibers undergo atrophy to a greater extent compared to type IIA fibers, irrespective of gender [33,34]. As for the hypoplasia, up to the age of 80 years, type II muscle fibers are the main fiber type lost in postural muscles [32], leading to an increase in the ratio of type I/II fibers; whereas, after 80 years of age, there is a similar decline in the number of type I and II fibers [35,36], which ultimately leads to a similar type I/II fiber ratio after the age of 85 years [37]. In addition, aging is associated with increased heterogeneity in MHC isoforms due to the preferential loss of fast motoneurons [38], which is thought to induce a shift towards the slower phenotype. This event, combined with the disuse-induced muscle atrophy, leads to the expression of fast MHC isoforms and physical activity able to reverse the shift towards a slower phenotype [39]. Thus, a distinctive MHC distribution in aged muscles does not exist, being the result of age-dependent neurodegenerative processes and physical activity status, which differ across individuals.

Another consequence of muscle mass loss is the increased susceptibility to injury and reduced capacity to regenerate [40]. This reduced capacity to regenerate has been linked to the loss of muscle satellite cells (MuSCs), the muscle stem cells, and metabolic reprogramming [41]. Available evidence suggests that the metabolite balance of both stem and differentiated cells can directly influence the epigenetic landscape through post-translational modifications of histone proteins, DNA sequence, and transcription factors [42,43]. Therefore, changes in metabolism may regulate many of the important cell fate decisions made by stem cells [44,45]. The impaired regeneration capacity of aged skeletal muscle has also been attributed to a decrease in the number of MuSCs, which after several cycles of regeneration/degeneration become exhausted and decline [46]. This decline in MuSC activity is accompanied by changes in the stem cell niche and/or cell-autonomous mechanisms such as oxidative damage [47]. Indeed, the MuSC transcriptome during aging is extensively reprogrammed, switching from genes involved in homeostasis to genes involved in tissue-specific stresses, such as DNA damage or inefficient autophagy [48].

Skeletal muscle metabolic remodeling during aging is associated with a reduction in the number of glycolytic fibers, reduced mitochondrial synthesis, and increased catabolism (red box). Decreased anaerobic glycolysis and impaired mitochondrial activity result in elevated protein catabolism and loss of muscle maintenance. Sarcopenic muscle shows reduced muscle fiber size (atrophy) and number (hypoplasia), and is accompanied by fat infiltration and connective tissue deposition. Among the main non-pharmacological approaches for the prevention of muscle mass loss during aging are long-term caloric restriction, dietary supplementation, and aerobic exercise (green box).

### 2.1. The Use of Specific Nutrients to Counteract Sarcopenia

An important aspect of aging is that the elderly consume less food compared with the young: they are less hungry and thirsty, eat smaller meals, and snack less [49]. Low food intake and monotonous diets put older people at risk of having malnutrition. Thus, in a vicious cycle, declining muscle strength and physical capability in older age may increase the risk of malnutrition, whilst malnutrition may contribute to further decline in physical capability [50]. Nonetheless, it is important to differentiate between malnutrition with deleterious consequences for skeletal muscle, and caloric restriction, which is a dietary regimen that reduces food intake without incurring malnutrition. Long-term caloric restriction has been demonstrated to significantly prevent the age-associated rewiring of the transcriptome [48]. However, in spite of its beneficial effects, caloric restriction is not enough to counteract aging, a phenomenon that is characterized by slow and progressive accumulation of epigenetic errors [51].

The nutrients that have been most consistently used to fight sarcopenia and frailty in older adults are vitamin D, proteins, and a number of antioxidant nutrients, that include carotenoids, selenium, and vitamins E and C [52]. Vitamin D has a genomic impact on skeletal muscle through the interference of the 1.25-VDR-RXR (retinoid receptor) heterodimer at certain nuclear receptors that influence gene transcription. Moreover, vitamin D has a non-genomic effect through a quick activation of intracellular signal transduction of 1.25 (OH) D to its non-nuclear receptors [53]. Interestingly, low serum levels of vitamin D are associated with an increased risk of sarcopenia in elderly adults [54]. Consistently, studies have reported that dietary supplementation with vitamin D improves muscle mass, function, and quality of life in sarcopenic elderly [55,56].

Another important nutrient for the elderly is, of course, protein. Dietary protein provides amino acids that are needed for the synthesis of muscle sarcomeric proteins. Importantly, absorbed amino acids have a stimulatory effect on muscle protein synthesis after feeding [57]. A blunted essential amino acid (EAA) response of muscle protein synthesis occurs in aged humans confined to bed rest, and it involves reduced mTORC1 signaling and amino acid transporter protein content. Supplementation with branched-chain amino acids (BCAA), and in particular leucine, plays an important role in the stimulation of postprandial muscle protein synthesis [58]. Nevertheless, protein turnover still decreases, even after supplementation with both types of amino acids, with continued loss of muscle mass.

Exploring possible treatments, it is necessary to mention EGb 761, a Ginkgo biloba extract, which has been shown to be an effective treatment in chronic age-dependent neurological disorders, by reversing the age-related metabolic shift from lipids to glucose utilization, promoting myogenesis, and restoring a more youthful gene expression pattern in sarcopenic rats [59].

Another possible treatment is represented by ghrelin. Ghrelin is a circulating peptidyl hormone mainly produced by the stomach, which, by acting on the hypothalamus and the pituitary, induces a strong release of GH and stimulates food intake and adiposity [60]. Elderly subjects have diminished levels of plasma ghrelin and these lower levels correlate with a decline in nutritional status [61]. Ghrelin exerts a strong and direct anti-atrophic activity on skeletal muscle and promotes skeletal muscle regeneration, showing anti-catabolic and anti-inflammatory effects, and leading to the inhibition of muscle protein catabolism [62,63]. Despite promising results in preclinical studies, further investigations are needed to clarify the benefits of the aforementioned specific nutrient supplementations on sarcopenia.

### 2.2. Exercise to Counteract Sarcopenia

Physical exercise, alongside a balanced diet, can contribute towards the prevention of muscle mass decline by increasing protein synthesis [17,64,65], inhibiting protein degradation, and stimulating muscle anabolism [65] (Figure 1). The cellular processes regulating the anabolic responses to exercise are more complex than those driven by nutrition alone. Thus, resistance-type exercise triggers multiple intramuscular signaling networks associated with cellular, biochemical, mechanical, and metabolic stress [64]. For example, while some authors have proposed a canonical pathway of regulation for mTORC1 via IGF1–PI3KAkt/PKB–mTOR, recent evidence points toward the existence of muscle intrinsic mechanosensitive signaling pathways that activate mTORC1 post exercise, such as the production of the lipid second messenger, phosphatidic acid (PA)/phospholipase D (PLD) [66,67], and adhesion proteins such as focal adhesion kinase (FAK) [68].

Skeletal muscle adaptation to aerobic exercise involves increased oxygen extraction and utilization [69], principally governed by mitochondrial capacity and function, which leads to increased biogenesis after 24 h of exercise [70]. As a proof of principle, lifelong aerobic exercise effectively reduces the biomarkers associated with sarcopenia in mice, preventing the aging-induced impairment of catabolic pathways and mitochondrial dysfunction, thereby counteracting and delaying aging-induced skeletal muscle atrophy [71]. However, similar to nutritional intake, the basal mitochondrial fractional synthesis rates decrease with aging [72], rendering the skeletal muscle adaptation to exercise less effective. Additional clinical research is needed to address the effectiveness of combined resistance and aerobic exercise protocols to counteract skeletal muscle sarcopenia.

## 3. Metabolic Alteration in Neurodegenerative Diseases

Neurodegenerative diseases, including amyotrophic lateral sclerosis (ALS), Alzheimer’s disease (AD), Parkinson’s disease (PD), and Huntington’s disease (HD), are characterized by progressive degeneration of specific neurons. Once neurodegeneration begins, the inexorable progression can only be slowed but not prevented. Protein aggregation, oxidative stress, and mitochondrial dysfunction are among some of the metabolic alterations observed in neurodegenerative diseases [73]. Whether these metabolic abnormalities play a direct role in the onset of these diseases or arise as a consequence of genetic and epigenetic factors is a matter of debate and remains to be elucidated.

### 3.1. Skeletal Muscle Metabolic Alterations in ALS

ALS is a neurodegenerative disease affecting the upper and lower motor neurons, resulting in muscle weakness, denervation, and eventual death [74]. Multiple mechanisms, including neuron excitability, glutamate toxicity, and protein aggregation, underlie the pathology of ALS. Cellular metabolic defects arise early and contribute to the clinical manifestation of the disease (Figure 2). Indeed, weight loss, dyslipidemia, and hypermetabolism are some of the metabolic alterations seen in affected patients (reviewed in [75,76]) and correlate with a worsening of the pathology. Despite this, the early events underlying the defective muscle metabolism in ALS as well as the major molecular mechanisms that cause ALS-associated hypermetabolism are still unknown. Indeed, it remains to be determined whether hypermetabolism is an epiphenomenon that might not be causally related to ALS progression, but rather, might simply be a result of muscle denervation.

Similar to human patients, the SOD1^G93A^ mouse model of ALS displays an increased energy expenditure as well as a decline in fat mass during the course of the disease [77,78]. Furthermore, the circulating level of fatty acids (NEFA) is not increased, suggesting that lipids that enter the circulation are rapidly utilized [77] (Figure 2). Since skeletal muscle in resting condition accounts for 20–30% of the total energy expenditure, it is reasonable to assert that alterations in the metabolic rate of this tissue may be crucial for disease progression.

It is interesting to note that denervation of glycolytic fibers is an early event in ALS muscles, suggesting that the metabolic signature of muscle fibers may be an important determinant of disease onset. As described for the mouse model of acute denervation, a reduction in glucose metabolism and GLUT4 expression and a switch to a lipid metabolism have been observed in asymptomatic ALS muscles [11,78,79,80]. This correlates with another early event in ALS, the increase in ROS as a consequence of elevated β-oxidation. In turn, ROS-mediated oxidative stress negatively impacts the insulin signaling cascade and alters the transcription of the glucose transporters responsible for basal and insulin-stimulated glucose uptake [81].

Although the loss of glycolytic fibers is a consequence of the degeneration of fast fatigable motor neuron contact that leads to a shift in muscle fiber metabolism, a cell autonomous effect has also been observed. Indeed, the dependency on fat oxidation is increased in ALS muscle cells, compared to healthy ones, suggesting a cell autonomous defect in energy metabolism [77]. In line with this observation, muscle-specific expression of mutated SOD1^G93A^ causes mitochondria abnormalities and dysfunction in vivo [82,83], parallel to neurovascular alterations, observed through X-ray-Phase-Contrast-Tomography (XPCT), at different stages of the disease [84]. XPCT demonstrated the ability to detect neuron alterations even in a pre-symptomatic state in SOD1^G93A^ mice. These mice display a shift from glycolytic toward slow oxidative fiber metabolism composition and a reduction in GLUT4 expression and glucose metabolism, resulting in muscle glycogen accumulation [83] (Figure 2).

Counteracting this event may represent a potential pharmacological strategy to delay disease progression. Indeed, fatty acid oxidation inhibition and switching to the use of glucose as fuel delay the onset and ameliorate disease progression [11,85].

It should be emphasized that metabolic alterations may differ during different stages of the disease. In addition, the different tissues involved (i.e., neurons, muscle, glial cells etc.) have different energy demands and substrates. This together with the heterogeneity of the pathology in different patients represents an area of research that deserves further investigation.

Among the metabolic pathways affected in ALS muscle are glucose and lipid metabolism, reactive oxygen species (ROS) production, and insulin signaling (red box). Reduced glucose transport and uptake, increased lipid metabolism, and ROS-mediated oxidative stress result in metabolic dysfunction. A high-fat, low-carb diet and physical exercise have been shown to have protective effects on neurons and skeletal muscles in ALS.

### 3.2. Dietary Intervention to Counteract ALS Progression

An increased body mass index (BMI) corresponds to an improvement in the lifespan of ALS patients, while a decreased BMI may increase disease severity [86,87,88,89], suggesting that dietary interventions can potentially alter the course of the disease. It is well known that lipolysis and fatty acid mobilization are a consequence of increased muscle energy requirements [90]; therefore, it is likely that the increased mobilization of lipids in an ALS mouse model occurs as a means to sustain a metabolic requirement in peripheral skeletal muscle [91]. Indeed, fatty acid metabolism provides more ATP compared to glucose metabolism [92], and Steyn and colleagues found that slower clinical decline in ALS patients was associated with higher fatty oxidation metabolism in myotubes [93]. Furthermore, a pilot study showed that a high caloric diet based on high carbohydrate content was well tolerated by ALS patients, and was associated with reduced serious adverse events [94].

This discovery is supported by a recent work showing that a high-fat diet is not only able to slow ALS progression but also to reduce the risk of developing ALS pathology [95,96], and a high level of circulating cholesterol and triglycerides is associated with prolonged survival of ALS patients [97]. Consistently, a high-fat diet improves motor function and increases survival of both TDP-43^A315T^ and SOD1^G93A^ ALS mouse models [98,99], supporting the idea that dietary supplementation may support hypermetabolic demand. Conversely, caloric restriction significantly reduced the survival of SOD1^G93A^ mice [100,101]. Mitochondria dysfunction and decreased activity of complex I are also associated with ALS pathology [102,103]. This observation suggests that a ketogenic diet (a high-fat, low-carb diet) may be optimal to support both hypermetabolic demand and to restore the function of complex I, promoting ATP synthesis. Indeed, a ketogenic diet prevents neuron loss and increases the lifespan of SOD1^G93A^ mice [104]. Similarly, a diet supplemented with caprylic triglyceride, as a source of ketone bodies, improves motor function in the same ALS mouse model [105].

In summary, available epidemiologic evidence supports the notion that malnutrition may contribute to ALS progression both in humans and in mice; conversely, increased fat and cholesterol intake might reduce the risk of ALS and the rate of disease progression (Figure 2). Based on the encouraging data from animal studies, there is now a great interest in pursuing dietary interventions for ALS. However, more clinical research is needed to explore the potential of the high-fat or ketogenic diet in the treatment of ALS.

### 3.3. Physical Exercise in ALS Progression

In addition to dietary interventions, physical exercise may also be an important modulator of muscle metabolism (Figure 2). Indeed, while low-intensity exercise induces a shift toward oxidative lipid metabolism, high-intensity exercise turns on the glycolytic metabolic pathway [106,107]. Moreover, exercise can promote the clearance of damaged mitochondria through the activation of the catabolic machinery [108]. Despite this evidence, the use of exercise as an adjunct therapeutic approach is still controversial. Indeed, epidemiologic data have revealed a higher incidence of ALS in subjects performing physical activities and athletes [109,110,111,112], while other studies have shown opposite results [113,114,115]. Without going into the merits of whether or not physical activity can facilitate the onset of ALS, we wish to explore the potential of exercise as a therapeutic intervention designed to slow disease progression.

Several clinical trials have demonstrated the benefits of exercise in ameliorating disease symptoms. Moderate exercise (i.e., 15 min performed twice daily) delays the deterioration of motor function and attenuates the decline in muscle strength [116,117,118]. Unfortunately, this beneficial effect is lost during disease progression [117]. While moderate exercise has been found to increase the lifespan of a SOD1^G93A^ mouse model of ALS [119], high-intensity exercise has no effect or is detrimental to the survival of these mice [120]. The quality of performed exercise seems to be critical for the beneficial effect observed in ALS. Thus, activation of the fast motor unit and the switching to glycolytic muscle metabolism exert beneficial effects in a mouse model of ALS [79]. However, in humans, research data are less clear. Overall, it seems that the best option to prevent the onset of fatigue and see improvements in ALS patients is an exercise of moderate intensity and not very high frequency, combining strength and aerobic endurance [121].

With the aim to clarify some of this discrepancy, Gerber et al. examined the effect of different running intensities on the survival of SOD1^G93A^ mice [120]. In contrast to previous studies, the authors used an appropriate control mouse, sedentary, and mice placed in a “sedentary treadmill” (treadmill powered off). Exercise did not improve mice survival at all the intensities analyzed (5, 10, and 21 cm/s, 15 min/day, 5 days/week). Nevertheless, increasing the intensity of exercise promoted the maintenance of body weight by increasing muscle mass [120].

Although useful, the results of these animal studies are difficult to reproduce in humans where the diagnosis of the disease is often made when the neurological damage is already significant. A weak or denervated muscle is more susceptible to overwork damage because it is already functioning close to its maximal limits. The hope is that in the future we will be able to design an appropriate exercise protocol, perhaps associated with an ideal diet, in order to maximize the chance of slowing down the pathology.

## 4. Skeletal Muscle Metabolism in Cancer Cachexia

Cancer cachexia is a multifactorial syndrome characterized by severe skeletal muscle wasting with or without adipose tissue loss, that develops irrespective of the given nutritional support [122]. It is a common feature of advanced cancer, caused by a combination of tumor- and host-derived factors, reduced food intake, and abnormal metabolism [123] (Figure 3).

Functional muscle mass is maintained by a dynamic balance between protein synthesis and degradation; a decrease in synthesis or an excessive degradation results in muscle wasting [124]. During tumor progression, the systemic network of anabolic and catabolic factors which regulates the balance between muscle synthesis and degradation is heavily compromised [125,126] (Figure 3). A decrease in circulating anabolic factors, such as insulin growth factor-1 (IGF-1) [127], leads to defective glucose handling and insulin resistance in tumor-bearing mice or cancer patients [128,129], overall contributing to muscle loss. A contemporaneous increase in circulating proinflammatory cytokines, which promote muscle catabolism, has been described in cachectic patients [130]. Moreover, tumor-derived extracellular vesicles (EVs) that promote myofiber death have been recently reported [131], further proving that a crosstalk between tumor and skeletal muscle contributes to the progression of cancer cachexia. Interestingly, besides signal transduction and cell communication, cell metabolism is one of the most common biological processes affected by tumor-derived EVs in cancer patients [132]. Additional studies will be necessary to determine the specific impact of EVs in mediating the crosstalk between tumor and skeletal muscle.

Recently, the involvement of neurogenic muscle atrophy was reported in a murine model of cancer cachexia [133], which may participate in muscle catabolism or metabolic reprogramming. Given that there is some controversy regarding this topic [134], further studies are needed to better define the involvement of compromised neuronal muscular junctions and muscle innervation in the progression of cancer cachexia.

Mitochondrial dysfunction associated with reduced ATP production [135,136,137,138] and elevated oxidative stress has been described in cachectic muscles [139,140,141,142]. Interestingly, mitochondrial dysfunction occurs before the onset of muscle wasting in tumor-bearing mice, suggesting a causative role in the activation of the proteolytic pathways [143]. In addition to the mitochondrial functional deficit, decreased mitochondriogenesis and altered mitochondria dynamics, probably due to high levels of circulating proinflammatory cytokines, contribute to altered muscle metabolism in cancer cachexia [144]. High resting energy expenditure is also typical of cancer patients [145], probably due to the increased expression of the uncoupling proteins UCP2 and UCP3 in cachectic muscles [146]. Dysfunctional mitochondria and compromised oxidative metabolism may, in turn, contribute to muscle insulin resistance and reduced protein metabolism [147,148,149].

While significant alteration in muscle metabolism has been extensively reported, metabolic reprogramming is not related to changes in fiber type composition, in cachectic patients [150,151,152] or in cancer cachexia murine models [153]. Instead, cachectic muscles show myosteatosis, i.e., accumulation of intramyocellular lipid droplets, both in humans and in rodents [154,155], which is associated with shorter survival [156]. Although abnormal expression of miRNAs [157], single nucleotide polymorphism [158], or alternatively spliced genes [159] associated with lipid metabolism have been reported, the mechanisms underpinning myosteatosis have not been fully elucidated.

In the red box are summarized the multiple pro-cachectic effects mediated by cancer. Impaired autophagy, energy expenditure, and denervation compromise protein homeostasis, leading to muscle wasting. In the green box are listed the anti-cachectic protective effects induced by exercise. Increased anti-inflammatory cytokines and suppression of the catabolic pathways lead to increased muscle mass and improved metabolism.

### 4.1. Nutritional Support to Counteract Cancer Cachexia

Muscle wasting correlates with poor prognosis in cancer patients, increasing morbidity and reducing both tolerance and responsiveness to treatments, ultimately accounting for up to 30% of deaths associated with cancer [160]. Importantly, preserving muscle mass extends survival in animal models of cancer cachexia [161,162]. Thus, interventions aimed to preserve functional muscle mass offer considerable potential in combination with anti-cancer regimens to enhance patient outcomes. Although nutritional support per se does not efficiently reverse the cachectic syndrome, it must be considered as an important component of a multi-targeted approach. To this purpose, nutritional therapies are under investigation as combined therapy, supporting a multimodal approach to counteract cachexia syndrome [163]. Anorexia and inadequate food intake occur in advanced cancer patients; therefore, dietary counseling and supplements are often suggested [164,165].

To improve muscle anabolism and help fight muscle wasting in cancer cachexia, the ketogenic diet characterized by increased protein intake has been proposed as a promising adjuvant treatment for cancer patients [166]. Although the optimal amount of proteins for muscle maintenance or gain has not yet been defined [167], a number of studies have suggested a dietary protein intake of >1.5 g/kg/day as a combined approach to counteract muscle wasting in cancer patients [168,169]. Interestingly, red and processed meat consumption is associated with higher risk of developing multiple cancers and with increased risk of overall cancer mortality [170], pointing to the importance of defining the optimal quantity, composition, and timing of protein intake.

Numerous mechanisms underlie the anti-tumor effects of ketogenic diets, including the reduced intake of glucose, which represents the primary metabolic fuel for many cancers because of the Warburg effect, and the promotion of ketone bodies synthesis, which reduce oxidative stress and exert anti-inflammatory action. Despite numerous studies and clinical trials demonstrating the potential of ketogenic diets [171], short- and long-term side effects of such rigid dietary regimens must be managed by a qualified nutritionist that should tailor the specific diet to individual cancer patients.

In addition to general nutritional strategies, numerous appetite stimulants such as megestrol acetate [172], l-carnitine [173], endocannabinoids [174], and ghrelin [175] have been the subject of recent clinical trials for their potential to increase food intake, thus promoting body weight gain and quality of life in cancer patients [176].

Establishing a net beneficial effect on skeletal muscle driven by nutrients without stimulating tumor growth, or negatively influencing anti-tumor therapy in cancer-related cachexia is a considerable challenge [177]. A number of dietary supplements, including omega-3 polyunsaturated fatty acids (N-_3_ PUFAs), polyphenols from fruits and tea, and vitamin antioxidants are known to exert anti-inflammatory effects in cancer conditions [178,179,180]. N-_3_ PUFAs, mainly sourced from marine fish and fish oil, have been reported to reduce cancer cachexia [181] as well as tumor growth [182,183], in addition to exerting numerous other health benefits in animals and humans [184,185].

Docosahexaenoic acid (DHA) and eicosapentaenoic acid (EPA) are the major effective PUFAs reported to improve body composition, muscle mass, and function [186,187] in cancer patients, and to suppress angiogenesis and cancer growth by multiple mechanisms [178,188]. Several lines of evidence also support a positive correlation between a higher intake of N-_3_ PUFAs and prolonged survival of cancer patients [178,189]. Previous studies have demonstrated a direct effect of N-_3_ PUFAs on muscle growth and hypertrophy. For instance, C2C12 myotubes supplemented with N-_3_ PUFAs show increased intracellular anabolic signaling with a concomitant increase in mitochondrial biogenesis, through enhanced expression of PPAR, PGC-1 α, carnitine palmitoyltransferase 1α and β (CPT1α; CPT1β), TFAM, and NRF genes [190,191]. Further studies confirmed that N-_3_ PUFAs increase muscle anabolic responses in human subjects [192,193,194,195], and reduce the catabolic pathways, lipid mobilization, and glucose consumption in cachectic patients, overall improving the body weight, lean body mass, and quality of life of cancer patients [178,196].

Among the polyphenols, epigallocatechin-3-gallate (EGCG), curcumin, resveratrol, and quercetin have been reported to reduce the catabolic pathways in cachectic muscle, by inhibiting the UPS pathway, whilst inducing IGF-1 anabolic signaling [197]. Moreover, resveratrol and quercetin are known inducers of SIRT1, thereby promoting mitochondrial biogenesis, preserving muscle oxidative capacity and increasing exercise tolerance [198,199,200], in addition to exerting anti-inflammatory, antioxidant, and anticarcinogenic effects through several mechanisms [201].

Several studies have proposed amino acid supplementation to improve cancer-induced muscle wasting by directly acting on muscle cells. In particular, a branched-chain amino acids (BCAAs) (i.e., valine, isoleucine, leucine)-enriched diet has been reported to counteract the negative protein turnover in skeletal muscle, without affecting tumor growth in cancer cachexia [202,203,204,205]. The positive effect of BCAAs on muscle protein turnover is mediated by multiple mechanisms, which remain to be fully clarified [206]. A recent study demonstrated that BCAAs drive increased phosphorylation of mTOR, 70 kDa ribosomal S6 kinase (*p*70^S6k^), and eIF4E-binding protein 1 (4E-BP1), parallel to the dephosphorylation of the eukaryotic initiation factor 2α (eIF2α), overall increasing protein synthesis while reducing protein degradation in MAC16 tumor-bearing mice [205].

A number of mechanisms by which vitamin D interferes with skeletal muscle function have been elucidated in muscle cell lines and in animal models, including modulation of muscle metabolism, by enhancing sensitivity to insulin, and mitochondria biogenesis and function [207,208]. Reduced circulating levels of vitamin D have been associated with impaired glucose metabolism and insulin sensitivity, in addition to a decrease in muscle mass, and performance in several pathologies [209,210,211,212]. In addition, vitamin D deficiency is highly prevalent in advanced cancer-related cachexia patients [213,214], highlighting the importance of vitamin D supplementation in this disease. However, contrasting activities of vitamin D have been reported in muscle cell lines and in tumor-bearing animals [180,213]. For instance, both pro- and anti-atrophic effects have been reported for two vitamin D metabolites on C2C12 myotubes upon addition of cancer cell-conditioned medium [215], discouraging the use of this treatment in vivo for rescuing cancer cachexia. Similarly, tumor-bearing rats treated with vitamin D displayed an impairment in the muscle regenerative program and any beneficial effect in counteracting muscle wasting [216]. In contrast, another study demonstrated vitamin D’s ability to reverse tumor-cell mediated changes in mitochondrial oxygen consumption and proteasomal activity in human skeletal muscle myoblasts [217], thus supporting the use of vitamin D to treat muscle wasting in cachexia. Differences in the experimental design (i.e., tumor-related cytokines in the conditioned medium) may account for the reported discrepancies regarding the effect of vitamin D on muscle wasting.

Overall, numerous studies support the use of dietary supplements to counteract muscle wasting in cancer cachexia; however, the effective doses, long-term effects, and potential side effects need to be further investigated to clearly determine the efficacy of these compounds. Moreover, additional experiments must investigate any potential effects of specific nutrients in conjunction with other pharmacological treatments, such as chemotherapy, on muscle homeostasis in cancer patients.

### 4.2. Exercise to Fight Cancer-Induced Cachexia

Physical exercise has been widely proposed as a promising intervention strategy for the prevention and treatment of cancer-related cachexia (Figure 3). In addition to favoring an increase in anti-inflammatory cytokines (i.e., IL-10, IL-1 receptor antagonist, and soluble TNF receptors-1 and-2) at the expense of pro-inflammatory cytokines, such as TWEAK, IL-1, INF-gamma, and LIF, among others [218,219], exercise negatively affects tumor mass growth and significantly improves muscle metabolism, highlighting the promising beneficial effects of exercise as a multitarget tool in the treatment of cancer cachexia [220,221,222]. Aerobic, resistance, and combined exercise training have been proven to be beneficial for cancer-induced muscle wasting, by modulating muscle mass and metabolism [223].

Exercise counteracts loss of muscle mass and functionality through suppression of catabolic pathways, whilst improving protein synthesis in cancer cachexia. Importantly, the amount of aerobic exercise directly correlates with the lifespan of tumor-bearing mice [162], highlighting the direct involvement of exercise in rescuing cancer-induced cachexia. Several studies have demonstrated exercise-driven modulation of the ubiquitin–proteosome system (UPS) and autophagy. Indeed, aerobic exercise reduces the gene expression level of the E3 ubiquitin ligases MuRF1 and Atrogin-1 in muscles of C26-bearing mice, thereby reducing muscle proteolysis [162,224]. Moreover, cachectic muscles are characterized by an accumulation of autophagic markers as a consequence of autophagic flux overloading [162,225]. Exercise triggers the autophagic flux [226], restoring Beclin-1 levels, including both isoforms of LC3b and p62, and ultimately leads to improved muscle homeostasis in tumor-bearing mice [162,223,224]. Similar results have been obtained administering pharmacological compounds known to trigger autophagy, i.e., AICAR and rapamycin, in tumor-bearing mice [162,227,228]. Moreover, a complete inhibition in the autophagic flux is deleterious in cancer cachexia, as proven by the early death of cachectic mice treated with colchicine [162], and further confirmed by increased muscle wasting promoted by knocking down Beclin-1 in cachectic muscles [229].

The anabolic resistance in cancer-induced muscle wasting seems to be attenuated by resistance exercise training through improvement in mammalian target of rapamycin (mTOR) signaling [230]. Indeed, muscle contraction triggers the expression of IGF-1 [231], which, in turn, activates several intracellular kinases, including phosphatidylinositol 3-kinase (PI3K), that promotes protein kinase B (Akt) activity, thereby promoting protein synthesis via the mTOR and glycogen synthase kinase 3β (GSK3β) kinases, in addition to counteracting protein degradation by suppressing the activation of the Forkhead (FoxO) transcription factors family.

In addition, mTOR and mitochondrial functions are strongly integrated. Indeed, mTOR modulates nuclear-encoded mitochondrial mRNAs and mitochondrial ribosomal protein translation through a 4E-BP1-dependent mechanism, thereby modulating mitochondrial mass, mitochondria biogenesis, and mitophagy [230]. Moreover, resistance exercise triggers the activity of other crucial genes responsible for mitochondrial biogenesis and dynamics, i.e., PGC1-α, NRF-1 and TFAM, overall enhancing mitochondria oxidative capacity and ATP production in different tumor-bearing animal models [224,232,233,234]. While suppressed PGC-1α expression in cachectic muscles correlates to disrupted mitochondrial dynamics and increased muscle wasting [144,235], discordant results have been reported on the effects of PGC-1α upregulation in skeletal muscle in Lewis lung carcinoma (LLC)-bearing mice. Thus, an increase in muscle mass was reported in Pin et al. [236], whilst cancer cachexia was not counteracted in Wang et al. [237] in PGC-1α overexpressing muscles LLC-bearing mice. Both studies reported an increase in tumor mass [236,237], highlighting the cross-talk between muscle and tumor and revealing a possible limitation of increasing the expression of PGC-1α in muscle in cancer cachexia [238]. In addition to improving mitochondrial oxidative capacity, exercise modulates muscle metabolism by enhancing GLUT-4 translocation at the sarcolemma [239], thereby increasing the productivity of the glycolytic pathway and counteracting insulin resistance [239,240,241].

In humans, several clinical trials have shown that physical activity during and after cancer treatment exerts numerous beneficial effects on functional, physical, and psychosocial outcomes, including muscle wasting and fatigue, metabolic and physical dysfunction, cognitive impairment, and depression [242,243,244,245]. To obtain health benefits, a combined protocol of intensive aerobic exercise of at least 75 min/week, or a moderate one of at least 150 min/week, with resistance training of major muscle groups 2–3 days per week for adults, has been recommended by the American College of Sports Medicine [246,247]. Alternatively, since cachectic patients may present physical limitations to performing exercise, several exercise mimetics are under investigation in preclinical trials to test their effectiveness to counteract dysmetabolism, overall preventing/delaying cachexia in cancer patients [248]. Among them, agonists of *P*PARδ or of AMPK, molecules able to activate SIRT1 or to reduce fatty acid oxidation, have given promising results to modulate lipid metabolism, mitochondrial function, and fiber-type determination in different muscle-wasting diseases, and cachexia as well [248,249]. Importantly, as part of a combined approach, exercise showed beneficial effects after chemotherapy in terms of reducing tumor growth rate, protein degradation and doxorubicin-induced toxicity in tumor-bearing mice [250].

## 5. Inter-or Transgenerational Effects of Nutrients and Exercise on Skeletal Muscle Mass

Parental nutritional support also influences future generations. Indeed, maternal low-protein diet [251,252] or high-fat diet (HFD) [253,254] consumption affect muscle metabolism and insulin sensitivity, predisposing the offspring to metabolic dysfunction later in life [255,256,257,258], in part by epigenetically altering skeletal muscle gene expression [259,260,261]. Moreover, maternal HFD correlates with compromised skeletal muscle development in the offspring, characterized by atrophy [253,262,263] and reduced performance [264,265], in part due to mitochondrial defects [263,266,267]. Even paternal nutrient support influences the metabolic state and the predisposition to metabolic disease [268,269], by epigenetic reprogramming of the sperm and offspring metabolic genes [270].

Importantly, maternal exercise can counteract the parental malnutrition-mediated effects in the offspring [271,272,273], by epigenetically modulating the expression of key regulators of skeletal muscle metabolism and mass, i.e., PGC-1α and the nuclear receptor Nr4a1 [271,274,275]. Interestingly, maternal exercise can even prevent the paternal obesity-induced metabolic dysfunction in the offspring skeletal muscle [276], in addition to paternal exercise [277].

Indeed, physical exercise is considered a physiological challenge whereby skeletal muscle has to remodel its metabolism, structure, and function in order to improve performance [278,279,280], via epigenetic regulation of gene transcription [281,282]. In addition, exercised muscles influence non-contractile tissues and whole-body metabolism by secreting myokines [283]. For these reasons, exercise is nowadays considered as a potential therapeutic approach for numerous diseases, including metabolic, oncological, and neurodegenerative diseases [13].

Recently, the beneficial effects of exercise have been shown to be transmitted to future generations, since parental physical exercise positively affects body metabolism, skeletal muscle structure, and performance of the offspring [263,273,284]. Studies from human and animal models demonstrated that maternal exercise performed during pregnancy increases muscle motility, reduces predisposition to obesity, and improves glucose homeostasis in the offspring [273,285,286,287,288]. Accumulating evidence indicates that paternal environmental exposure and physiology impact offspring development. In addition to increased sperm parameters, such as viability and motility [289,290], paternal exercise has been associated with increased mass and altered expression of metabolic genes in the offspring’s skeletal muscle [291,292].

Therefore, numerous studies have highlighted the important role of nutrients and parental physical exercise in the epigenetic regulation of offspring muscle metabolism. Additional studies are needed to clarify the long-term effects of parental exercise on offspring muscle metabolism and performance in humans. New advanced techniques, such as scanning X-ray micro-diffraction [293], can be exploited in the future to provide structural details on the skeletal muscle fibers.

## 6. Conclusions

In conclusion, available epidemiologic evidence supports the notion that many deleterious effects of physio-pathological conditions, such as sarcopenia, ALS, or cancer-induced cachexia, can be prevented by specific nutritional support or physical exercise. Moreover, acting on metabolic cues, through dietary interventions or physical exercise, may also be beneficial for the epigenetic regulation of muscle metabolism in the offspring. Therefore, nutritional support and exercise can be proposed as nonpharmacological approaches to ameliorate skeletal muscle homeostasis in numerous pathological states, by regulating protein synthesis, degradation, and metabolism. Despite encouraging data from animal studies supporting these alternative approaches, more clinical research is necessary to validate the potential of specific dietary treatments or exercise protocols.

## Figures and Tables

**Figure 1 metabolites-11-00517-f001:**
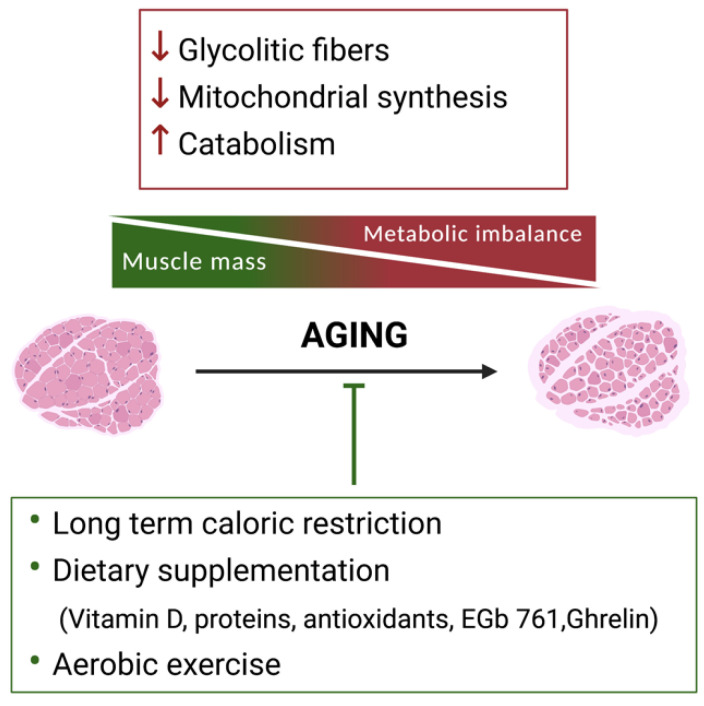
Metabolic reprogramming of skeletal muscle during aging and non-pharmacological approaches to counteract sarcopenia. Skeletal muscle metabolic remodeling during aging is associated with a reduction in the number of glycolytic fibers, reduced mitochondrial synthesis, and increased catabolism (red box). Decreased anaerobic glycolysis and impaired mitochondrial activity result in elevated protein catabolism and loss of muscle maintenance. Sarcopenic muscle shows reduced muscle fiber size (atrophy) and number (hypoplasia), and is accompanied by fat infiltration and connective tissue deposition. Among the main non-pharmacological approaches for the prevention of muscle mass loss during aging are long-term caloric restriction, dietary supplementation, and aerobic exercise (green box).

**Figure 2 metabolites-11-00517-f002:**
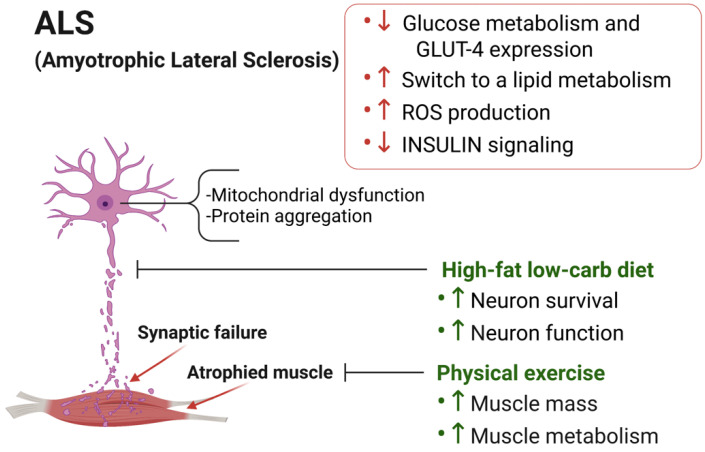
Skeletal muscle metabolic reprogramming and proposed approaches to counteract disease progression. Among the metabolic pathways affected in ALS muscle are glucose and lipid metabolism, reactive oxygen species (ROS) production and insulin signaling (red box). Reduced glucose transport and uptake, increased lipid metabolism, and ROS-mediated oxidative stress result in metabolic dysfunction. A high-fat-low-carb diet and physical exercise have been shown to have protective effects on neurons and skeletal muscles in ALS.

**Figure 3 metabolites-11-00517-f003:**
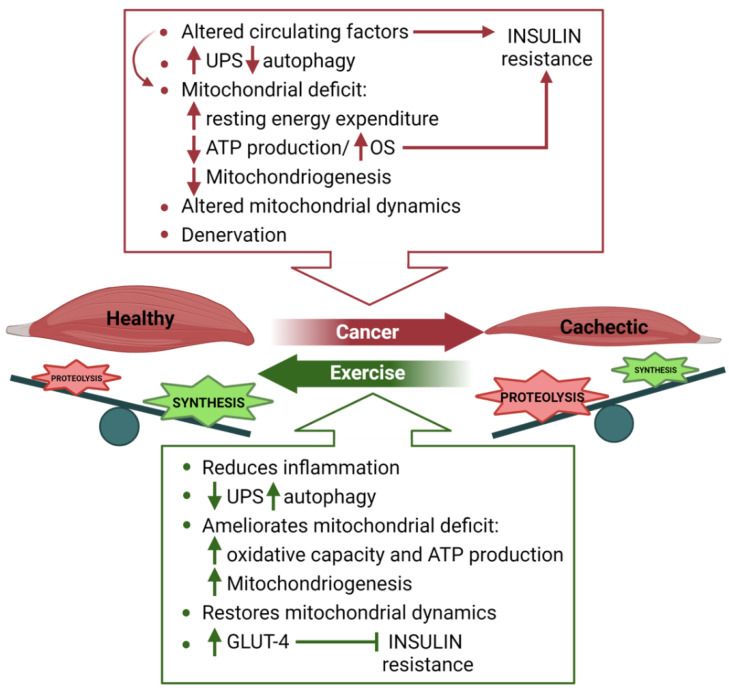
Metabolic reprogramming in cancer-induced cachexia and beneficial effects of exercise. In the red box are summarized the multiple pro-cachectic effects mediated by cancer. Impaired autophagy, energy expenditure, and denervation result in unbalanced protein homeostasis, overall leading to muscle wasting. In the green box are listed the anti-cachectic protective effects induced by exercise. Increased anti-inflammatory cytokines and suppression of the catabolic pathways lead to increased muscle mass and improved metabolism.

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
