# Peer review of "Metabolic Remodeling in Skeletal Muscle Atrophy as a Therapeutic Target"

_metabolites, 2021, doi:10.3390/metabo11080517_

Round 1

Reviewer 1 Report

The review of Renzini et al covers the changes in the metabolism in muscles during aging and in conditions of ALS and cancer cachexia. Particular emphasis the authors make on the impact of diet and physical exercises on the muscle fibers metabolism in these pathological conditions. The review discusses a very important and attractive opportunity of usage of specific nutrients and physical exercises to combat muscular decline associated with aging, ALS, or cancer. First, the authors describe the metabolism remodeling in muscles in general and after provide the reader with a comprehensive review on the topic in each specific condition. The review is of particular interest for a broad scientific audience as well as for medical researches. However, I have some minor concerns/suggestions to the authors:

  1. The introduction part and the aging paragraph provide very brief information regarding metabolic types of muscle fibers. The authors should consider the addition of details to this part as this information is crucial for the review and is mentioned literally in every paragraph.
  2. The figure legends should be added; without the description, the figures are not neat.
  3. The change in muscle fibers in figure 1 is a bit confusing. What exactly happens with the fibers so they look different.
  4. Figure 2 signature “mitochondrial dysfunction” pointing to the nucleus. It is confusing. The authors should improve figures to make them more clear. Figure 3 contain the whole loop of events that are not commented on in legend

Author Response

We thank the reviewer for the positive comments.

Please find attached our point-by-point response.

Reviewer 2 Report

The manuscript by Renzini and colleagues is a well written and up to date review of the state of the art in muscle metabolic abnormalities associated with muscle wasting. The simple and well-designed figures usefully support the text content.

My only major concern regards the choice of adopting ALS to be compared with aging sarcopenia and cachexia, that is only in part understandable. I think muscle dystrophies in general would better compare to cachexia and sarcopenia, both being multifactorial and including a broad pathogenetic spectrum. In the case the authors want in any case focus on ALS, I suggest to add an introduction to the chapter summarizing the main neurodegenerative diseases along with the metabolic defects and then deepen ALS description as in the present version.

Minor:

Figure1, change syntesis to synthesis;

The reference to Figure1 appears very late in the text after the figure. The same applies for the following ones.

As for cancer cachexia, the authors say that ‘develops irrespective of the given nutritional support’, however in the following chapter they propose nutritional intervention as a viable option.

Author Response

We are grateful for the reviewer’s comments.

Please find attached our point-by-point response.

Reviewer 3 Report

I congratulate Renzini and colleagues for a generally well written review on targeting of muscle remodeling for skeletal muscle atrophy.  I enjoyed reading their review and believe that other scientists and researchers will also find value in reading this updated, topical and somewhat critical review of this area.  I have some minor suggestions for improvement:

General:

  1. Writing: Please go over the writing in fine detail to identify grammatical and spelling errors:

e.g. Line 81, p.3 “…”changes, consisting in of an increase of anaerobic glycolysis…”

e.g. Line 304, p.8 “…A decrease in circulating anabolic factors, such 304 as insulin growth-like factor-1 (IGF-1)

  1. Figure 1 has numerous spelling issues, please fix

Main text

  1. Line 80, p.3 “during aging muscles are subjected to metabolic changes, consisting in an increase of anaerobic glycolysis leading to reduced oxygen availability and energy production (Figure 1). As a consequence, there is a loss of type II glycolytic fibers and clustering of type I oxidative fibers”

This part is a bit confusing. Sarcopenic changes do involve a more preferential reduction type II fibres, however those energetic changes noted appear to suggest the opposite change

  1. Line 141, p.4: “Given the importance of oxidative stress and inflammation in the reduction of protein synthesis, antioxidant and anti-inflammatory compounds, such as polyphenols, have 142 been extensively studied for their health benefits [50]. Nevertheless, further evidence is 143 needed to establish their therapeutic potential in inflammatory conditions [41].”

This paragraph is a bit brief and vague.  Consider if you need this here or perhaps provide greater detail on specific antioxidant/anti-inflammatory compounds found to target protein turnover or metabolic remodelling.

  1. In the discussion on ALS, the statement on line 222, p.6 states: “Counteracting this event may represent a potential pharmacological strategy to delay disease progression. Indeed, fatty acid oxidation inhibition or alternative glucose oxidation boosting delay the onset and ameliorate disease progression [10].”

However, in the subsequent paragraphs on dietary interventions to counteract ALS progression, a high fat diet as well as a high carbohydrate diet were found to be effective at slowing ALS progression. The previous argument seems a bit counterintuitive to what evidence is suggesting here. 

  1. When discussing “3.2. Physical exercise in ALS progression”, it might be worth clarifying and delineating evidence from clinical trials using aerobic vs resistance training protocols. It is a bit unclear what “moderate exercise” has entailed in these studies.

The final paragraph in that section raises a good point, although probably should be referenced.

  1. When describing “4.1. Nutritional support to counteract cancer cachexia” it is perhaps a bit odd that you haven’t included general nutritional strategies such as higher energy and protein intakes, which are arguably more impactful than supplements. Nutritional interventions alone don’t appear potent enough to combat advanced cancer cachexia.
  2. When describing “4.2. Exercise to counteract cancer cachexia”, potential mechanisms of exercise have been well explained, however there is a relative lack of discussion on patient-important physical outcomes (except for the final paragraph of that section). I think it is worth discussing if available evidence permits it here.

Author Response

We thank the Reviewer for the constructive comments. 

Please find attached our point-by-point response.
